# Exposure to 16 Hz Pulsed Electromagnetic Fields Protect the Structural Integrity of Primary Cilia and Associated TGF-β Signaling in Osteoprogenitor Cells Harmed by Cigarette Smoke

**DOI:** 10.3390/ijms22137036

**Published:** 2021-06-29

**Authors:** Yangmengfan Chen, Romina H. Aspera-Werz, Maximilian M. Menger, Karsten Falldorf, Michael Ronniger, Christina Stacke, Tina Histing, Andreas K. Nussler, Sabrina Ehnert

**Affiliations:** 1Siegfried Weller Research Institute at the BG Trauma Center Tübingen, Department of Trauma and Reconstructive Surgery, University of Tübingen, Schnarrenbergstr. 95, D-72076 Tübingen, Germany; chenyangmengfantj@163.com (Y.C.); rominaaspera@hotmail.com (R.H.A.-W.); mmenger@bgu-tuebingen.de (M.M.M.); thisting@bgu-tuebingen.de (T.H.); sabrina.ehnert@med.uni-tuebingen.de (S.E.); 2Sachtleben GmbH, Haus Spectrum am UKE, Martinistraße 64, D-20251 Hamburg, Germany; falldorf@citresearch.de (K.F.); ronniger@citresearch.de (M.R.); stacke@citresearch.de (C.S.)

**Keywords:** extremely low frequency pulsed electromagnetic fields (ELF-PEMFs), mesenchymal stem cells, bone healing, cigarette smoke extract, TGF-β signaling, primary cilium

## Abstract

Cigarette smoking (CS) is one of the main factors related to avoidable diseases and death across the world. Cigarette smoke consists of numerous toxic compounds that contribute to the development of osteoporosis and fracture nonunion. Exposure to pulsed electromagnetic fields (PEMF) was proven to be a safe and effective therapy to support bone fracture healing. The aims of this study were to investigate if extremely low frequency (ELF-) PEMFs may be beneficial to treat CS-related bone disease, and which effect the duration of the exposure has. In this study, immortalized human mesenchymal stem cells (SCP-1 cells) impaired by 5% cigarette smoke extract (CSE) were exposed to ELF-PEMFs (16 Hz) with daily exposure ranging from 7 min to 90 min. Cell viability, adhesion, and spreading were evaluated by Sulforhodamine B, Calcein-AM staining, and Phalloidin-TRITC/Hoechst 33342 staining. A migration assay kit was used to determine cell migration. Changes in TGF-β signaling were evaluated with an adenoviral Smad2/3 reporter assay, RT-PCR, and Western blot. The structure and distribution of primary cilia were analyzed with immunofluorescent staining. Our data indicate that 30 min daily exposure to a specific ELF-PEMF most effectively promoted cell viability, enhanced cell adhesion and spreading, accelerated migration, and protected TGF-β signaling from CSE-induced harm. In summary, the current results provide evidence that ELF-PEMF can be used to support early bone healing in patients who smoke.

## 1. Introduction

Cigarette smoking (CS) is the principal cause of avoidable disease and death across the world in the 21st century. According to the project summary of Special Eurobarometer 458–Attitudes of Europeans towards tobacco and electronic cigarettes (https://ec.europa.eu/health/tobacco/eurobarometers_en accessed on 3 February 2021.), 23% of EU residents are defined as daily smokers. The percentage of daily smokers is especially high for southern European countries, e.g., Bulgaria (38%), Greece (42%), or Croatia (35%). Cigarette smoke extract (CSE) consists of more than 6,000 molecular species and toxic compounds. When applied to cells, CSE was proven to inhibit various signaling pathways, i.e., transform growth factor β (TGF-β) [1], insulin-like growth factor 1 [2], phosphoinositide 3-kinase (PI3K), and protein kinase B [3], as well as antioxidative enzyme activities [4]. Moreover, CS severely affects the skeletal system, contributing to the development of osteoporosis [5], delayed fracture healing, and nonunion formation [6], the treatment of which adds a tremendous burden to the medical expenditure and socio-economy [7].

Along with the growing awareness of the negative effects of CS on the skeletal system, there is an increasing demand for effective therapies. In the recent decades, a variety of mechanical stimuli were used to treat skeletal diseases by maintaining the balance between osteoblastic bone formation and osteoclastic bone resorption [8]. Since 1978, pulsed electromagnetic fields (PEMFs) were recognized as a safe, non-invasive, and effective physical treatment for bone diseases, thus received substantial attention in the field of regenerative medicine [9]. Under physiological conditions, PEMFs will be generated in the human body due to piezoelectric phenomena [10]. For example, the frequency of mechanical strains caused by muscle activity is in the extremely low frequency (ELF) range between 5 Hz and 30 Hz [11]. Theoretically, this allows ELF-PEMFs to mimic the physiological stimulation, and thus to become a feasible treatment for the musculoskeletal system. Indeed, screening ELF-PEMFs with frequencies of up to 100 Hz revealed that osteoprogenitor cells respond well to the mentioned frequency range below 30 Hz [12]. Besides frequency, which has been investigated in earlier studies, there might be other parameters responsible for the type and magnitude of PEMF effects in cells and tissue such as intensity and exposure timing (duration and intervals). The latter was examined in this study. Nevertheless, research to date has not yet revealed the detailed mechanisms of how cells sense, translate, and transduce signals of ELF-PEMFs to regulate cellular activities.

To explore a potential mechanism, we focused on the primary cilium that is considered as the “mechanosensing antenna” in nearly all eukaryotic cells [13]. In bone tissue, the microtubule-based non-motile primary cilium protrudes from the membrane of osteoblasts. It touches the inner wall of the ‘lacunae’ to sense extracellular mechanical forces exerting on the bone tissue [14]. In response to mechanical signals, numerous receptors are recruited to the ciliary membrane to facilitate the activation of associated signaling pathways [15]. As a result, extracellular physical signals are sensed. For patients who smoke daily, the structural integrity of primary cilia, and thus the mechanosensory ability of bone cells, are damaged. This can be simulated in vitro by applying CSE to the cells [13]. The canonical TGF-β signaling, which plays a crucial role in early fracture healing [16], is strongly inter-dependent on the structural and functional integrity of the primary cilia [1,17,18,19]. TGF-β is known to promote bone regeneration, as it can enhance viability, proliferation, and migration of osteoprogenitor cells [20,21,22]. In smokers, not only do TGF-β levels fail to increase after trauma or orthopedic surgery [23,24], but also the primary cilia-mediated TGF-β signaling is inhibited [1]. This is proposed to impair the healing process. In this study, we intended to assess possible protective effects of ELF-PEMFs on cell viability, adhesion, and migration of immortalized human mesenchymal stem cells (SCP-1 cells) affected by CSE. Furthermore, we assessed whether the effects relate to the structure of the primary cilia and the associated TGF-β signaling.

## 2. Results

### 2.1. ELF-PEMFs Increased Viability and Alleviated the Negative Effects of CSE in SCP-1 Cells

Cell viability was determined in SCP-1 cells exposed to 16 Hz ELF-PEMFs for 0, 7, 30, and 90 min/day in the presence or absence of 5% CSE. As expected, CSE significantly impaired cell viability (Calcein-AM staining) and decreased cell numbers (total protein content). Exposure to ELF-PEMFs increased cell viability, both in the presence or absence of the CSE. The strongest effects were observed with 30 min daily exposure to the ELF-PEMF (Figure 1A,B).

### 2.2. SCP-1 Cell Adhesion and Spreading Are Enhanced by ELF-PEMF Exposure

Cell adhesion and spreading are important for initiating the biological function of osteoprogenitor cells. In this study, microscopic visualization of the cytoskeleton and the nuclei was used to observe cell adhesion and spreading when exposed to 16 Hz ELF-PEMFs and/or 5% CSE (Figure 2A). Quantification of the number of adherent nuclei indicated that CSE impaired cell adhesion, but 7 min and 30 min daily exposure to 16 Hz ELF-PEMF favored cell adhesion (Figure 2B). Similarly, the mean cellular size indicated that 7 min and 30 min daily exposure to 16 Hz ELF-PEMF facilitated cell spreading, both in the presence or absence of 5% CSE (Figure 2C).

### 2.3. Exposure to ELF-PEMFs Enhances Migration of SCP-1 Cells

Migration of osteoprogenitor cells plays a crucial role in early fracture healing. To evaluate the influences of 16 Hz ELF-PEMFs on SCP-1 cell migration, a cell migration assay kit was used to generate a cell-free circular area (migration zone) centrally in the cavities of 96-well plates (Figure 3A). Automated image evaluation demonstrated that 5% CSE strongly suppressed SCP-1 cell migration, while 16 Hz ELF-PEMFs accelerated SCP-1 cell migration both in the presence or absence of 5% CSE. The strongest effects of the 16 Hz ELF-PEMF were observed with 30 min daily exposure (Figure 3B).

Overall the positive effect of the 16 Hz ELF-PEMF on SCP-1 cell viability, adhesion, spreading, and migration also displayed a positive effect on the osteogenic differentiation (Appendix A). In line with earlier publications [12,25], daily exposure to 16 Hz ELF-PEMF induced alkaline phosphatase (ALP) activity and formation of mineralized matrix in SCP-1 cell. Prolonging the duration of the daily exposure from 7 min to 30 min, but not to 90 min, further enhanced the positive effect of the 16 Hz ELF-PEMF on the SCP-1 cells. The 30 min daily exposure to the 16 Hz ELF-PEMF was sufficient to reverse the well-known negative effect that continuous stimulation with 5% CSE has on SCP-1 cells [1,4,13].

### 2.4. ELF-PEMF Exposure Intensified TGF-β Signaling in SCP-1 Cells

TGF-β is a key factor favoring the migration of osteoprogenitor cells [22]. As exposure to 16 Hz ELF-PEMF induced SCP-1 cell migration, potential alterations in TGF-β signaling were investigated. Functional, canonical (Smad2/3) TGF-β signaling was quantified using an adenoviral reporter assay. As expected, medium supplementation with 5 ng/mL recombinant human TGF-β activated Smad2/3 signaling in SCP-1 cells. The presence of 5% CSE significantly reduced this effect. Exposure to the 16 Hz ELF-PEMFs intensified the TGF-β signaling by 50 to 60%, both in the presence or absence of 5% CSE. Significant results were observed again for 30 min daily exposure with the 16 Hz ELF-PEMF (Figure 4A); therefore, this condition was used for all further experiments.

Interestingly, a Western blot revealed that the amount of total Smad2 and Smad3 phosphorylated was only affected in the presence of 5% CSE and not by exposure to the 16 Hz ELF-PEMF. However, the basal levels of Smad2 and Smad3 were significantly reduced in the presence of 5% CSE. A period of 30 min exposure to 16 Hz ELF-PEMF significantly increased cellular levels of Smad2 and Smad3 (Figure 4B,C)

As expected, SCP-1 cell migration was accelerated in presence of 5 ng/mL TGF-β. The addition of TGF-β alone was not able to restore the SCP-1 cell migration impaired by 5% CSE. However, an additional 30 min daily exposure to 16 Hz ELF-PEMF fortified the positive effect of TGF-β, to fully restore the SCP-1 cell migration suppressed by 5% CSE (Figure 4D).

### 2.5. ELF-PEMF Exposure Reversed CSE-Mediated Changes in Smad2, -3, and -7 Expression

In line with the Western blot results, gene expression of both *Smad2* and *Smad3* was significantly reduced in SCP-1 cells exposed to 5% CSE. The 30 min daily exposure to 16 Hz ELF-PEMF did not significantly affect the basal expression levels of *Smad2* and *Smad3*, but restored their expression in presence of 5% CSE (Figure 5A–C). As a regulator for TGF-β signaling, the expression levels of Smad7 were also determined. SCP-1 cells exposed to 5% CSE showed higher expression of the inhibitory *Smad7*. The 30 min daily exposure to 16 Hz ELF-PEMF did not alter the basal expression level of *Smad7*, but significantly reduced *Smad7* expression in presence of 5% CSE (Figure 5D).

### 2.6. The Structural Integrity of Primary Cilia Is Related to the Protective Effects of 16 Hz ELF-PEMFs

The primary cilium functions as the “antenna” in cells that can sense extracellular physical stimuli, e.g., ELF-PEMFs. During osteogenic differentiation, SCP-1 cells form primary cilia structures. The morphology and distribution of primary cilia were determined by the immunofluorescence staining (Figure 6A). Image analysis revealed that supplementing the differentiation medium with 5% CSE caused a significant drop in the amount of ciliated SCP-1 cells and a significant decrease in the average length of the primary cilia. A 30 min daily exposure for 5 days to 16 Hz ELF-PEMF significantly increased the number of ciliated cells and the length of the primary cilia, both in the presence or absence of the CSE (Figure 6B,C).

## 3. Discussion

TGF-β is a key factor regulating the early phases of bone regeneration. The absence of the rise in TGF-β levels following a fracture is associated with a delayed or even impaired healing process [16]. It has been reported that smokers lack the rise in TGF-β levels expected following a fracture or orthopedic surgery [23,24]. Furthermore, CS was reported to harm the structural integrity of primary cilia by increasing oxidative stress [13], which in turn affects TGF-β signaling [1,18,19]. The data presented here provide evidence that exposure to 16 Hz ELF-PEMFs protects the structural integrity of the primary cilia and thus rescues TGF-β signaling in osteoprogenitor cells harmed by CSE. This way, 16 Hz ELF-PEMFs may support the early phases of bone regeneration. This is supported by a rodent study showing impaired fracture healing in mice with non-functional primary cilia [26]. In addition, a recent study showed that exposure to sinusoidal EMFs protected the rise in primary cilia structure and thus enhanced bone mineral density in rats, an effect attributed to the activation of BMP and Wnt signaling [27].

For almost half a decade, PEMFs were used in clinical settings to support bone regeneration. Addressing a complex repair process involving many components acting at different phases or time points, it has been a challenge to optimize PEMF exposure parameters (i.e., frequency of field, intensity of field, timing, and duration). Consequently, the physical parameters of the applied PEMFs varied on a large scale [9]. Despite being recognized as a safe and non-invasive physical treatment for bone diseases, the application of PEMFs could not yet be fully established in the clinical routine [28]. The present study utilized ELF-PEMFs with a fundamental frequency of 16 Hz (more details given in the Materials and Methods section). A screening approach identified ELF-PEMFs in this frequency range to best support the viability and differentiation of osteoprogenitor cells [12]. The effectiveness of the ELF-PEMF was confirmed in further studies, both in vitro and in vivo [25,29]. Comparing the effect on bone formation in this in vivo study [29] with other studies that used 4- to 200-times longer daily exposures revealed that increasing the exposure duration may enhance the observed effects [9]. Thus, in the present study, exposure duration was varied at first. Increasing the daily exposure from 7 min to 30 min could further improve viability, adhesion, spreading, and migration of SCP-1 cells. However, further elongation of the daily exposure duration reduced the observed positive effect of 16 Hz ELF-PEMF exposure on SCP-1 cells. It has been reported that repetitive exposure to this specific 16 Hz ELF-PEMF, induced anti-oxidative defense mechanisms in osteoprogenitor cells by formation of reactive oxygen species (ROS) [25]. It is possible that prolonged exposure to the 16 Hz ELF-PEMF causes an accumulation of ROS that becomes detrimental when reaching a certain level.

This is of special interest when this ELF-PEMF shall be used to support bone healing in patients with an already increased oxidative stress level, e.g., smokers. In vitro exposure to CSE significantly increases ROS levels in osteoprogenitor cells, affecting the cells’ viability and function [4]. The viability, adhesion, spreading, and migration of osteoprogenitor cells are crucial for initiating bone regeneration after injury. Our data show that exposure to 5% CSE, which simulates smoking 10 cigarettes per day, significantly reduced the viability, adhesion, and migration of SCP-1 cells. Exposure to the 16 Hz ELF-PEMF improved the viability, adhesion, spreading, and migration of CSE-exposed SCP-1 cells. Daily exposure of 7 min (16 Hz ELF-PEMF) was sufficient to abrogate the damaging effects of the CSE exposure, when compared to control conditions. Daily exposure of 30 min (16 Hz ELF-PEMF) further ameliorated the effect, reaching levels above the control cells. Comparing the 16 Hz ELF-PEMF effects between the CSE and control group revealed that a longer duration of exposure was more effective in the control group. This might be another hint that prolonging the exposure time can cause accumulation of ROS, which more quickly reaches a threshold under stress conditions, e.g., with CSE.

As a possible regulatory mechanism, how CSE inhibits osteogenic function, CSE-dependent destruction of the cells’ primary cilia was described [13], which impairs TGF-β signal transduction [1]. The importance of primary cilia for TGF-β signal transduction is highlighted with an in vivo study, which showed impaired fracture healing in mice lacking functional primary cilia in osteoprogenitor cells [30]. However, there are reports suggesting that PEMF therapy requires functional primary cilia on osteogenic cells to exert their effects [31,32]. The primary cilium is covered by many receptors required for initiating cell signaling [33,34]. PEMF exposure not only increases expression of such receptors, but also initiates their localization at the base of the primary cilium [31]. The pocket surrounding the base and proximal part of the primary cilium is a site for clathrin-dependent endocytosis, reported to regulate TGF-β signaling. The localization of the TGF-β receptors to the endocytotic vessels was reported to activate Smad2/3 and ERK1/2 (extracellular signal-regulated kinases 1 and 2) signaling at the ciliary base [18]. Our data show that SCP-1 cells with CSE-damaged primary cilia still respond to the 16 Hz ELF-PEMF exposure. It is described that primary cilia adapt their length in response to mechanical stimuli, which affects the mechanosensitivity of the cells. While the loss of primary cilia integrity is associated with reduced cell signaling [1,33], the lengthening of the microtubuli is reported to enhance cellular mechanosensitivity [35]. Our data revealed that exposure to 16 Hz ELF-PEMF elongated the primary cilia structures both in the presence or absence of the CSE. Therefore, enhanced cell signaling following the physical stimuli is expected.

TGF-β is robustly released from the bone matrix immediately after the bone fracture happens [36]. A temporal increase in TGF-β serum levels following a fracture is associated with successful fracture healing [16]. It has been reported that smokers lack this increase in TGF-β serum levels following a fracture or orthopedic surgery [23,24], which is thought to contribute to the delayed fracture healing in these patients. Thus, there is evidence that smokers with a fracture not only lack the required increase in TGF-β serum levels, but also that their osteoprogenitor cells may not adequately respond to the growth factor. In our experiment, exposure to CSE decreased the expression of *Smad2* and *Smad3*. Consequently, less Smad2 and Smad3 were phosphorylated. Exposure to CSE significantly lowered the ratio of phosphorylated Smad3 to Smad3, suggesting that CSE inhibits TGF-β signaling not only by inhibiting the expression of the regulator Smads, but also upstream before the phosphorylation of Smad3. The latter could be related to the structural integrity of the primary cilia, which regulate clathrin-dependent endocytosis reported to promote TGF-β-induced activation of Smads and transcriptional responses [37,38].

Several studies described that exposure to PEMFs significantly increases the expression of TGF-β in osteogenic cells both in vitro and in vivo [39]. However, little is known about the underlying mechanisms. One study proposed that increased expression of TGF-β in PEMF-treated MSCs is facilitated by an increase in miRNA-21, which suppresses the expression of the inhibitory *Smad7* [40]. This could explain our findings, which show decreased expression of *Smad7* in SCP-1 cells exposed to 16 Hz ELF-PEMFs. Other well-reported target genes of miRNA-21 include *TGFβR*, the PI3K inhibitor *PTEN*, and the ERK1/2 inhibitor *Spry* [41]. The latter is in accordance with our finding that the positive effect of 16 Hz ELF-PEMF stimulation on osteoprogenitor cells requires activation of ERK1/2 signaling [12].

## 4. Materials and Methods

If not cited differently, chemicals, media and medium supplements were purchased from Sigma-Aldrich, which is now part of Merck (Darmstadt, GER).

### 4.1. Cell Culture

The human-immortalized mesenchymal stem cell line SCP-1 was used in this study. SCP-1 cells were provided by Professor Matthias Schieker. SCP-1 cells were cultured in α-MEM medium (Gibco, Darmstadt, Germany) supplemented with 5% fetal bovine serum (FBS) in a water-saturated atmosphere of 5% CO_2_ at 37 °C. For osteogenic differentiation, SCP-1 cells at a confluence of 90% were cultured in an osteogenic differentiation medium (α-MEM medium supplemented with 1% FBS, 200 μM L-ascorbate-2-phosphate, 5 mM β-glycerol-phosphate, 25 mM HEPES, 1.5 mM CaCl_2_, and 100 nM dexamethasone). The culture medium was changed twice a week.

### 4.2. ELF-PEMF Device and Exposure

The ELF-PEMF devices (Somagen^®^), provided by the Sachtleben GmbH (Hamburg, Germany), are medical devices certified according to European law (CE 0482, compliant with EN ISO 13485:2016 + EN ISO 14971:2012). The ELF-PEMFs applied in this study have a fundamental frequency of 16 Hz and an intensity of 6-282 µT (B field amplitude 6 mm above the applicator), which is emitted as groups of pulses (bursts) in sending–pause intervals [12]. The daily ELF-PEMF exposure was 7, 30, or 90 min.

The device applicators distort the local earth magnetic field due to a magnetic foil in the panels. This leads to an inhomogeneous local earth magnetic field (DC), which superpositions with the alternating magnetic field (AC) of the applicator coil. The AC field corresponds here to the ELF-PEMF. The magnetic field is defined by
(1)$${\vec{B}}_{DC}={\left(B_{DC,x},B_{DC,y},B_{DC,z}\right)}^{\tau }$$
(2)$${\vec{B}}_{AC}\left(\vec{r},t\right)={\left(B_{AC,x}\left(\vec{r},t\right),B_{AC,y}\left(\vec{r},t\right),B_{AC,z}\left(\vec{r},t\right)\right)}^{\tau }$$
which leads to the total magnetic field
(3)$${\vec{B}}_{total}\left(\vec{r},t\right)={\vec{B}}_{DC}+{\vec{B}}_{AC}\left(\vec{r},t\right),$$
which is finally exposed to the cells.

The AC magnetic field of the applicator coil shows a common dipole characteristic and vanishes very fast with distance (Figure 7).

### 4.3. Cigarette Smoke Extract (CSE) Preparation

To generate the CSE, the smoke of cigarettes (Marlboro, Philip Morris, New York, NY, USA) was bubbled through a plain α-MEM culture medium at a speed of 100 bubbles/min. The concentration of the CSE solution was evaluated photometrically at 320 nm. An optical density (OD) of 0.7 was considered as 100% CSE. After being filtered and mixed with the culture medium, 5% CSE (which is considered as smoking 10 cigarettes per day) was used in this study [42].

### 4.4. Cell Viability

Cell viability was quantified by sulforhodamine B (SRB) staining, which was used to determine the total protein content. In detail, SCP-1 cells were fixed with ice-cold 99% ethanol (–20 °C) for 1 h. Subsequently, cells were washed 3 times with PBS and stained with SRB staining solution (0.4% SRB in 1% acetic acid, Sigma-Aldrich, Munich, Germany) for 30 min at room temperature. Then, cells were washed 4 to 5 times with 1% acetic acid to remove unbound SRB. For quantification, the bound SRB was resolved with 10 mM unbuffered TRIS solution (pH~10.5). The OD was measured with a microplate reader at 565–690 nm [4].

### 4.5. Live Staining and Cytoskeleton Staining

Calcein-AM staining was performed to determine cell viability. Briefly, cells were washed 3 times with PBS and subsequently incubated with Calcein-AM (2 μM) and Hoechst 33,342 (2 μg/mL) in plain medium at 37 °C for 30 min. Images of cells were taken with a fluorescence microscope (EVOS FL, life technologies, Darmstadt, Germany). To explore cell adhesion and morphological changes, the cytoskeleton of the SCP-1 cells was stained with phalloidin-TRITC. Cells were fixed with 4% formaldehyde at room temperature for 10 min, actin cytoskeleton was stained with phalloidin-TRITC (2 ng/mL), and nuclei were counter-stained with Hoechst 33,342 (2 ng/mL) for 30 min at room temperature. After washing with PBS, images were taken with a fluorescence microscope and quantified with the ImageJ software [13].

### 4.6. Cell Migration Assay

Cell migration was evaluated by using the cell migration assay kit (Tebu-bio, Offenbach, Germany). Sterilized stoppers were placed in 96-well plates. Then, SCP-1 cells were seeded around the inserts at a concentration of 4 × 10^4^ cells/well in a growth medium. After 24 h, the stoppers were removed from the wells, and the cells were washed 3 times with PBS. Then, the growth medium was replaced by an osteogenic differentiation medium with or without stimuli (ELF-PEMF, CSE, and TGF-β). Immediately, an image was taken with the microscope to document time point 0. After 72 h, SCP-1 cells were stained with SRB for better visualization. Microscopic images were analyzed with the ImageJ software and the “gap closure” was calculated.

### 4.7. Immunofluorescent (IF) Staining for Primary Cilia

SCP-1 cells were fixed with a 4% formaldehyde solution for 10 min at room temperature, then incubated with 0.2% Triton-X-100 for 10 min, and subsequently with 2% formaldehyde for 10 min. After that, cells were blocked with 5% bovine serum albumin (BSA, Carl Roth, Darmstadt, Germany) for 1 h. Then, cells were incubated with primary antibody solution (acetylated α-tubulin, 1:100, Santa Cruz, Heidelberg, Germany) overnight at 4 °C. The next day, cells were washed with PBS and incubated with Alexa-488 labeled secondary antibody (1:2000, Invitrogen, Karlsruhe, Germany) and Hoechst 33,342 (2 ng/mL) for 2 h. Fluorescent images were taken with a fluorescence microscope (200- or 400-fold magnification) and analyzed with the ImageJ software [13].

### 4.8. Transient Cells Infection and Reporter Assay

Adenoviral reporter constructs (Smad2/3 reporter, Ad5-CAGA9-MLP-Luc, 1:10 *v/v*) [1] were used to infect SCP-1 cells. After 24 h, cells were washed with PBS, and treated with or without 5% CSE (in the osteogenic differentiation medium) for 72 h. After 24 h, cells were additionally stimulated with 10 ng/mL recombinant human TGF-β1 for 48 h. Finally, cells were lysed with a lysis buffer (Luciferase substrate kit, Promega, Madison, WI, USA); 20 µL cell lysate was mixed with 20 µL luciferase substrate, and the luminescent signal was detected immediately with the microplate reader. The luminescence was normalized to the total protein content, determined by micro Lowry [43].

### 4.9. Conventional RT-PCR

Reverse transcription polymerase chain reaction (RT-PCR) was used to evaluate the expression of genes involved in the TGF-β signaling pathway. In brief, RNA was isolated by Phenol–Chloroform extraction. The cDNA synthesis kit (Fermentas, St. Leo-Rot, Germany) was used to synthesize the first-strand cDNA. Primer sequences and PCR parameters are summarized in Table 1. The products of RT-PCR were subjected to a 2% (*w*/*v*) agarose gel electrophoresis with ethidium bromide, and the results were analyzed by using ImageJ software.

### 4.10. Western Blot

To harvest total protein, cells were lysed in ice-cold RIPA buffer (50 mM TRIS, 250 mM NaCl, 2% NP40, 2.5 mM EDTA, 0.1% SDS, 0.5% DOC, and protease/phosphatase inhibitors: 1 μg/mL pepstatin, 5 μg/mL leupeptin, 1 mM PMSF, 5 mM NaF, and 1 mM Na_3_VO_4_) and the lysate was centrifuged (3000× *g*, 10 min) to remove cell debris. The protein concentration was measured by micro Lowry [43]; 25 μg of total protein were separated by SDS-PAGE (10% acrylamide-bisacrylamide gels, 100 V, 180 min) and subsequently transferred to nitrocellulose membranes (100 mA, 180 min). Protein separation and transfer were checked by Ponceau staining; 5% BSA was used to block the unspecific binding sites. Membranes were incubated with primary antibodies against p-Smad2, p-Smad3, Smad2/3 (all 1:1000, sc-133098, sc-517575 from Santa Cruz Biotechnology, Heidelberg, Germany and 3108 from Cell Signaling Technologies, Danvers, MA, USA), and GAPDH (1:5000, G9545 from Sigma-Aldrich, Munich, Germany) diluted in TBST, overnight at 4 °C. Subsequently, membranes were incubated with the corresponding HRP-labeled secondary antibodies (1:10,000 in TBST) for 2 h at room temperature. Membranes were covered with enhanced chemiluminescent substrate solution (1.25 mM luminol, 0.2 mM p-coumaric acid, 0.03% H_2_O_2_ in 100 mM TRIS, pH = 8.5) and a CCD camera was used to detect the chemiluminescent signals. Signal intensities were determined with the ImageJ software.

### 4.11. Statistical Analysis

Results are displayed as mean ± SEM. Data were analyzed with two-way ANOVA followed by Tukey’s multiple comparison test using the GraphPad Prism software (El Camino Real, CA, USA); *p* < 0.05 was considered as statistically significant.

## 5. Conclusions

In line with the literature, our data show that 5% CSE, which is equivalent to smoking 10 cigarettes per day, negatively affects the viability, adhesion, spreading, migration, and differentiation of SCP-1 cells. Daily exposure to 16 Hz ELF-PEMFs partly reversed the negative effect of the CSE, especially when the duration of the daily exposure was increased from 7 min to 30 min, but not longer. As a possible regulatory mechanism, an ELF-PEMF-mediated rescue of the primary cilia structure and the amount of ciliated cells was identified. This, in turn, was associated with a fortification of the canonical (Smad2/3) TGF-β signaling and a decreased expression of *Smad7*. In summary, our data suggest that 30 min daily exposure to the 16 Hz ELF-PEMF can be used as an adjunct therapy to support early fracture healing in smokers, who frequently have to struggle with delayed or impaired fracture healing.

## Figures and Tables

**Figure 1 ijms-22-07036-f001:**
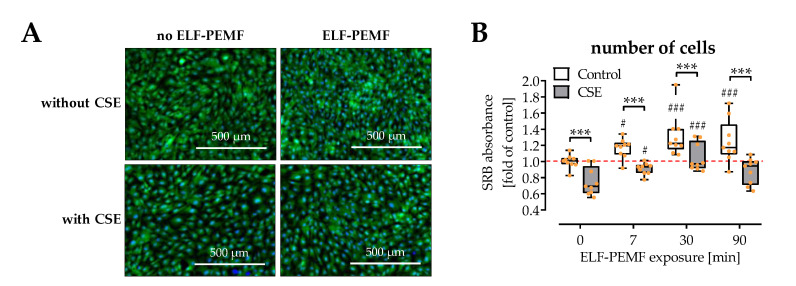
SCP-1 cell viability is affected by 16 Hz ELF-PEMFs (0, 7, 30, and 90 min) and 5% cigarette smoke extract (CSE). (**A**) Fluorescent Calcein-AM staining (2 µM, green) was used to visualize living cells, and Hoechst 33,342 (2 μg/mL, blue) was used as nuclear counterstain. (**B**) Sulforhodamine B (SRB) staining was used to quantify cell numbers by total protein content after 3 days. N = 3, *n* = 3. Data are presented as a box plot (Min to Max with single data points). Data were compared by non-parametric two-way ANOVA followed by Tukey’s multiple comparison test: *** *p* < 0.001 as indicated; ^#^ *p* < 0.05 ^###^ *p* < 0.001 as compared to the respective control (no ELF-PEMF).

**Figure 2 ijms-22-07036-f002:**
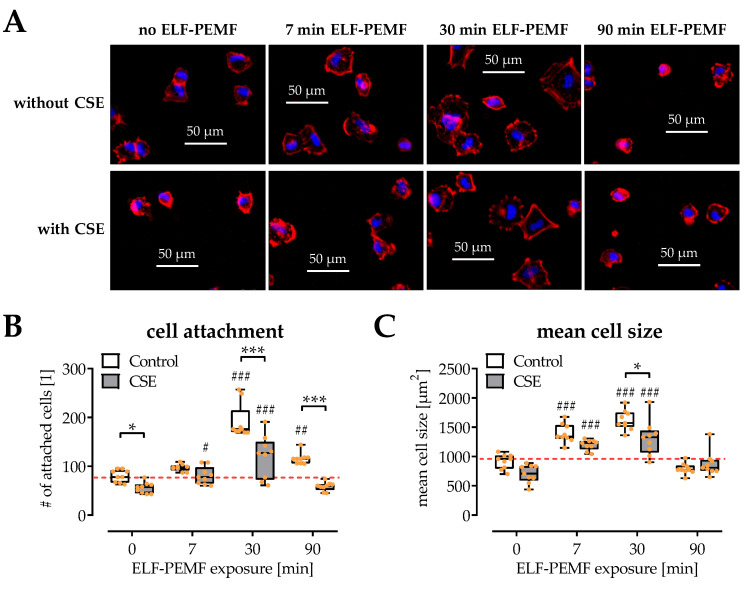
Influences of 16 Hz ELF-PEMFs (0, 7, 30, and 90 min) and 5% cigarette smoke extract (CSE) on SCP-1 cell adhesion and spreading after 4 h. (**A**) Representative images (400× magnification) of the fluorescence staining for cytoskeleton (phalloidin-TRITC, 2 μg/mL, red) and nuclei (Hoechst 33342, 2 μg/mL, blue). (**B**) Automated quantification of adherent nuclei, and (**C**) the mean size of attached SCP-1 cells using the ImageJ software. N = 3, *n* = 3. Data are presented as box plots (Min to Max with single data points). Data were compared by non-parametric two-way ANOVA followed by Tukey’s multiple comparison test: * *p* < 0.05 and *** *p* < 0.001 as indicated; ^#^ *p* < 0.05, ^##^ *p* < 0.01, and ^###^ *p* < 0.001 as compared to the respective control (no ELF-PEMF).

**Figure 3 ijms-22-07036-f003:**
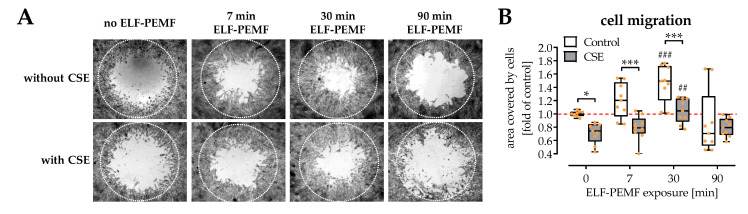
Migration of SCP-1 cells is affected by 16 Hz ELF-PEMFs (0, 7, 30, and 90 min) and 5% cigarette smoke extract (CSE). (**A**) Sulforhodamine B (SRB) staining was performed to better visualize cells invading into the migration zone (dotted circle) after 72 h. (**B**) SCP-1 cell migration was quantified using the ImageJ software. N = 4, *n* = 2. Data are presented as a box plot (Min to Max with single data points). Data were compared by non-parametric two-way ANOVA followed by Tukey’s multiple comparison test: * *p* < 0.05 and *** *p* < 0.001 as indicated; ^##^ *p* < 0.01 and ^###^ *p* < 0.001 as compared to the respective control (no ELF-PEMF).

**Figure 4 ijms-22-07036-f004:**
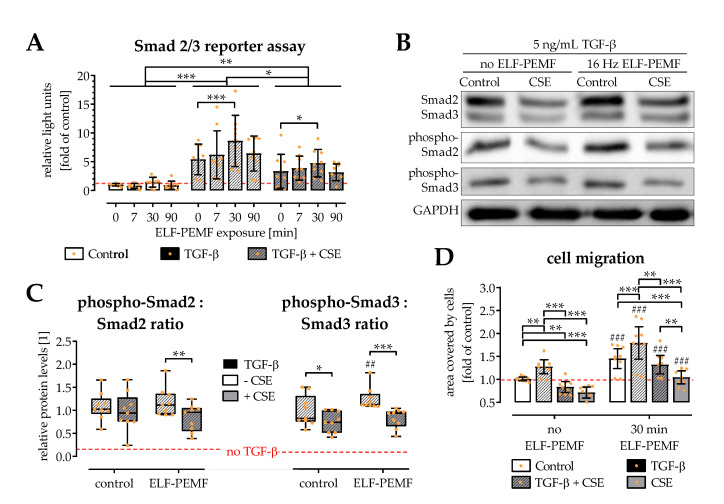
Canonical (Smad2/3) TGF-β signaling affected by 5% cigarette smoke extract (CSE) is fortified by exposure to the 16 Hz ELF-PEMFs. (**A**) An adenoviral reporter assay was used to quantify canonical (Smad2/3) TGF-β signaling in SCP-1 cells exposed to 5 ng/mL TGF-β, 5% CSE, and/or the 16 Hz ELF-PEMFs (0, 7, 30, 90 min daily exposure) for 72 h. Western blot was used to confirm phosphorylation of Smad2 and Smad3 in the cells with 30 min daily exposure to the 16 Hz ELF-PEMF. (**B**) Representative image of the Western blot. (**C**) Signal intensities were quantified with the ImageJ software and the ratio of phosphorylated-Smad2 to Smad2 and phosphorylated-Smad3 to Smad3 were determined. (**D**) SCP-1 cell migration was determined using the cell migration assay kit. Cells invading the migration zone were quantified using the ImageJ software. N = 3, *n* = 3. Data are presented as box plots (Min to Max with single data points). Data were compared by non-parametric two-way ANOVA followed by Tukey’s multiple comparison test: * *p* < 0.05, ** *p* < 0.01, and *** *p* < 0.001 as indicated; ^##^ *p* < 0.01 and ^###^ *p* < 0.001 marking the ELF-PEMF effect.

**Figure 5 ijms-22-07036-f005:**
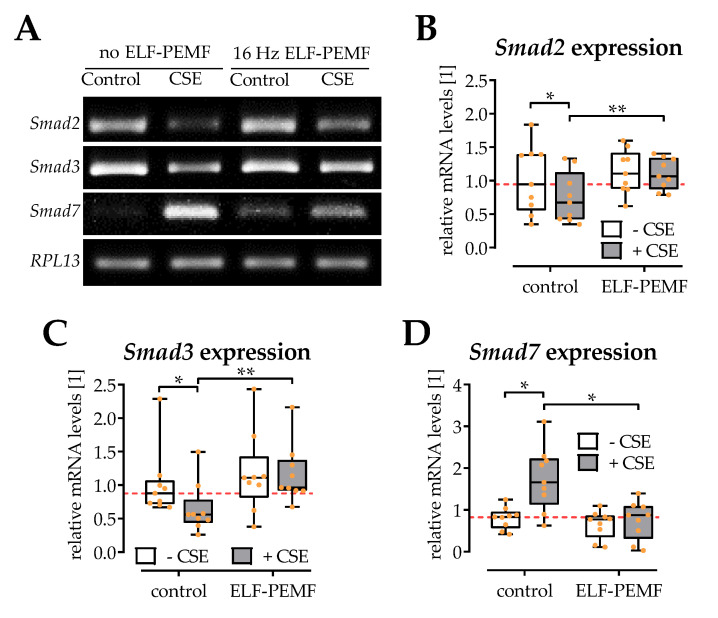
Gene expression of *Smads* is affected by exposure to 5% cigarette smoke extract (CSE) and 16 Hz ELF-PEMFs for 30 min daily for 3 days. (**A**) Representative RT-PCR images. Signal intensities were quantified with the ImageJ software. Expression of (**B**) *Smad2*, (**C**) *Smad3*, and (**D**) *Smad7* was normalized to *RPL13* (house-keeping gene). N = 3, *n* = 3. Data are presented as box plots (Min to Max with single data points). Data were compared by non-parametric two-way ANOVA followed by Tukey’s multiple comparison test: * *p* < 0.05 and ** *p* < 0.01 as indicated.

**Figure 6 ijms-22-07036-f006:**
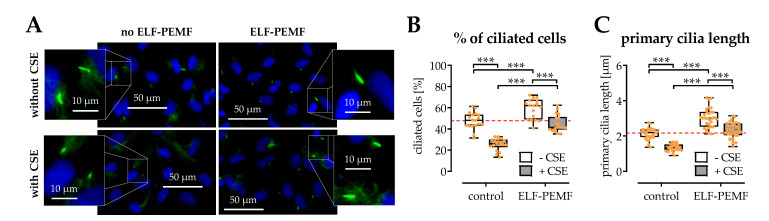
The structural integrity of primary cilia is affected by exposure to 5% cigarette smoke extract (CSE) and rescued by exposure to 16 Hz ELF-PEMFs for 5 days and 30 min daily. (**A**) Primary cilia were visualized by immunofluorescence staining for acetylated tubulin (green). Nuclei were counterstained with Hoechst 33,342 (blue). Fluorescent images were analyzed with the ImageJ software, to determined (**B**) the amount of ciliated cells in percent and (**C**) the length of the primary cilia. N = 3, *n* = 3. Data are presented as box plots (Min to Max with single data points). Data were compared by non-parametric two-way ANOVA followed by Tukey’s multiple comparison test: *** *p* < 0.001 as indicated.

**Figure 7 ijms-22-07036-f007:**
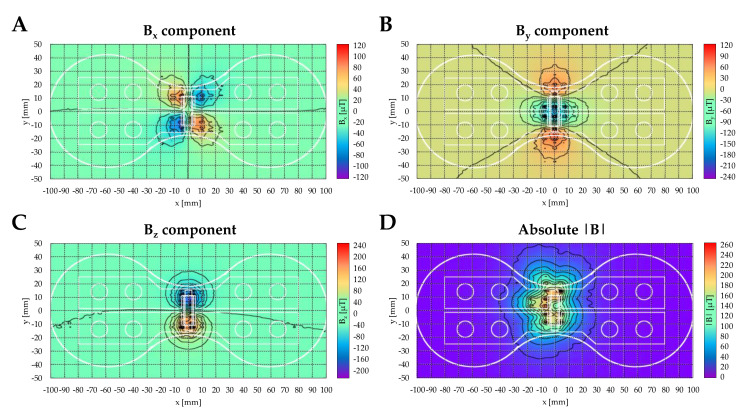
The AC magnetic field distribution of the applicator coil for all 3 directions and the absolute field approximately 4 mm above the coil. Heat maps summarize the distribution of the magnetic flux density: (**A**) B_x_ component, (**B**) B_y_ component, (**C**) B_z_ component, and (**D**) absolute |B|. The measurement was taken with the recalibrated 3 axes PNI RM3100 sensor for a constant current (I = 0.1 A) after subtracting the local earth magnetic field (I = 0 A).

**Table 1 ijms-22-07036-t001:** Primer information.

Target Gene	Gene Bank Accession Number	Forward Primer (5′–3′)	Reverse Primer (5′–3′)	Amplicon Size [bp]	Ta [°C]	Number of Cycles
***Smad2***	NM_001003652.3	CAAACCAGGTCTCTTGATGG	GAGGCGGAAGTTCTGTTAGG	259	60	35
***Smad3***	NM_005902.2	GGAGAAATGGTGCGAGAAGG	GAAGGCGAACTCACACAGC	259	60	35
***Smad7***	NM_005904.1	TTCGGACAACAAGAGTCAGC	AAGCCTTGATGGAGAAACC	200	60	35
***RPL13***	NM_012423.3	AAGTACCAGGCAGTGACAG	CCTGTTTCCGTAGCCTCATG	100	56	30

## Data Availability

The datasets generated during and/or analyzed during the current study are available from the corresponding author on reasonable request.

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
