# Peer review of "Exposure to 16 Hz Pulsed Electromagnetic Fields Protect the Structural Integrity of Primary Cilia and Associated TGF-β Signaling in Osteoprogenitor Cells Harmed by Cigarette Smoke"

_ijms, 2021, doi:10.3390/ijms22137036_

Round 1
Reviewer 1 Report
The study by Chen et al. examine the effects of pulsed electromagnetic fields (PEMFs) on osteoprogenitor cells exposed to cigarette smoke extract (CSE). The authors used SCP-1 cells and determine their viability, TGF-b signaling, and primary cilia following PEMFs and CSE. The authors conclude that PEMF can rescue SCP-1 cells from CSE induced harm.
Cigarette smoking is a chronic condition that is detrimental to a wide range of human health, including bone and fracture healing. The authors' study design however does not address this particular clinical significance. 1) Acute exposure to CSE already affects viability of SCP-1 cells (within 30 minutes). This questions the study design where it is really appropriate to conclude the effects of chronic cigarette smoking on bone health. 2) SCP-1 cells were never induced to osteogenic differentiation. The outcome assays are limited to general characteristics of SCP-1 cells, such as viability, adhesion, migration, etc, but not osteogenic differentiation. To really support authors' conclusion that "PEMF can be used to support early bone healing in patients that smoke", these two major concerns should be addressed.
While the paper is easy to read, there are several instances of spelling/grammar mistakes that should be corrected prior to publication (for instance see lines 83, 202, 205 to name a few).
Minor comments:
1) In all the figures, authors should show the individual data points on their box-whisker plots.
2) Please describe the cell density/confluence for SCP-1 cells when induced for CSE/PEMF exposure.
Author Response
Please find the detailed answeres to the reviewers comments in the attached file.

Reviewer 2 Report
The authors were aimed to assess possible protective effects of ELF-PEMFs on cell viability, adhesion, and migration of immortalized human mesenchymal stem cells (SCP-1 cells) affected by Cigarette smoke extract (CSE). Furthermore, was observed the effects relate to the primary cilia structure associated TGF-β signaling (key factor regulating the early Their data show that exposure to 16 Hz ELF-PEMFs improves viability, adhesion, spreading, and migration of SCP-1 cells, both in the presence or absence of 5% CSE.
The study covers some issues that have been overlooked in other similar topics. The study was conducted with a good scientifically sound. The structure of the manuscript appears adequate and well divided in the sub-paragraphs.
Issues that need improvement: Please check English grammar and typos thorough the text.
Conclusion Section: This paragraph required a general revision to eliminate redundant sentences and to add some "take-home" message.
Author Response

(The authors gave the same response as above.)

Round 2
Reviewer 1 Report
Authors have appropriately answered concerns raised by the reviewers. However, please explain how the sample size is N=3 when there are more than 3 dots in the figures.
Author Response
Thank you very much for carefully looking at the changes made to the manuscript. In order to show reproducibility and validity of the assays, we included single data points for both biological replicates (N = 3) and technical replicates (n = 3 or higher for the image analysis) in the plots.